# Antiplatelet Effects of Selected Xanthine-Based Adenosine A_2A_ and A_2B_ Receptor Antagonists Determined in Rat Blood

**DOI:** 10.3390/ijms241713378

**Published:** 2023-08-29

**Authors:** Monika Kubacka, Szczepan Mogilski, Marek Bednarski, Krzysztof Pociecha, Artur Świerczek, Noemi Nicosia, Jakub Schabikowski, Michał Załuski, Grażyna Chłoń-Rzepa, Jörg Hockemeyer, Christa E. Müller, Katarzyna Kieć-Kononowicz, Magdalena Kotańska

**Affiliations:** 1Department of Pharmacodynamics, Jagiellonian University Medical College, 9 Medyczna Street, 30-688 Krakow, Poland; monika.kubacka@uj.edu.pl (M.K.); szczepan.mogilski@uj.edu.pl (S.M.); 2Department of Pharmacological Screening, Jagiellonian University Medical College, 9 Medyczna Street, 30-688 Krakow, Poland; marek.bednarski@uj.edu.pl (M.B.); nicosia.noemi@hsr.it (N.N.); 3Department of Pharmacokinetics and Physical Pharmacy, Jagiellonian University Medical College, 9 Medyczna Street, 30-688 Krakow, Poland; krzysztof.pociecha@uj.edu.pl (K.P.); artur.swierczek@uj.edu.pl (A.Ś.); 4Division of Neuroscience, Vita Salute San Raffaele University, 20132 Milan, Italy; 5Department of Technology and Biotechnology of Drugs, Faculty of Pharmacy, Jagiellonian University Medical College, 9 Medyczna Street, 30-688 Krakow, Poland; j.schabikowski@gmail.com (J.S.); zaluski.michal@gmail.com (M.Z.); katarzyna.kiec-kononowicz@uj.edu.pl (K.K.-K.); 6Department of Medicinal Chemistry, Faculty of Pharmacy, Jagiellonian University Medical College, 9 Medyczna Street, 30-688 Kraków, Poland; grazyna.chlon-rzepa@uj.edu.pl; 7PharmaCenter Bonn, Pharmaceutical Institute, Pharmaceutical & Medicinal Chemistry, An der Immenburg 4, D-53121 Bonn, Germany; joerg.hockemeyer@uni-bonn.de (J.H.); christa.mueller@uni-bonn.de (C.E.M.)

**Keywords:** adenosine A_2A_, A_2B_ receptors, adenosine A_2A_, A_2B_ receptor antagonists, anti-aggregation effect, antiplatelet activity, phosphodiesterase activity, lipid peroxidation

## Abstract

The platelet aggregation inhibitory activity of selected xanthine-based adenosine A_2A_ and A_2B_ receptor antagonists was investigated, and attempts were made to explain the observed effects. The selective A_2B_ receptor antagonist PSB-603 and the A_2A_ receptor antagonist TB-42 inhibited platelet aggregation induced by collagen or ADP. In addition to adenosine receptor blockade, the compounds were found to act as moderately potent non-selective inhibitors of phosphodiesterases (PDEs). TB-42 showed the highest inhibitory activity against PDE3A along with moderate activity against PDE2A and PDE5A. The antiplatelet activity of PSB-603 and TB-42 may be due to inhibition of PDEs, which induces an increase in cAMP and/or cGMP concentrations in platelets. The xanthine-based adenosine receptor antagonists were found to be non-cytotoxic for platelets. Some of the compounds showed anti-oxidative properties reducing lipid peroxidation. These results may provide a basis for the future development of multi-target xanthine derivatives for the treatment of inflammation and atherosclerosis and the prevention of heart infarction and stroke.

## 1. Introduction

Adenosine is an important purine metabolite with effects on cardiovascular physiology and pathology [1]. In particular, adenosine affects platelet aggregation, which plays a critical role in hemostasis and thrombosis [2]. Research to date underscores the pivotal role of adenosine and its receptors and ligands, particularly agonists and antagonists of adenosine A_2A_ and A_2B_ receptors, in modulating platelet aggregation [3]. Adenosine receptors represent a subfamily of G protein-coupled receptors, commonly expressed in membranes of various types of tissues and cells; however, the distribution of subtypes is highly tissue-specific [4]. Notably, blood platelets express only two (A_2A_ and A_2B_) of the four known receptor subtypes (A_1_, A_2A_, A_2B_, and A_3_). Among these, the adenosine A_2A_ receptor shows a higher affinity for adenosine and is expressed in higher density on platelets as compared with the A_2B_ receptor [5].

Early studies identified the crucial role of adenosine A_2A_ and A_2B_ receptors in inhibiting platelet aggregation via the activation of the receptors, which results in an increase in intracellular cyclic adenosine monophosphate (cAMP), and a decrease in intracellular calcium levels [6]. Therefore, adenosine A_2A_ and A_2B_ receptors may be considered targets for antiplatelet therapy. Apart from adenosine, a range of synthetic, selective, and longer-lasting agonists of adenosine A_2A_ and A_2B_ receptors has shown platelet-inhibiting properties and usefulness in preventing thrombus formation. Also, moderately or non-selective compounds such as NECA, HE-NECA, CGS-21680 [4], and recently, PSB-15826 [7], have been reported in the literature to possess antithrombotic properties.

In contrast, adenosine A_2A_ and A_2B_ receptor antagonists may enhance platelet aggregation by counteracting the action of endogenous adenosine. For instance, the inhibitory effect of cilostazol on platelet aggregation in whole blood was significantly reversed with ZM241385, an adenosine A_2A_/A_2B_ receptor antagonist [8]. Also, adenosine A_2A_ receptor knock-out mice demonstrated an increase in platelet aggregation [3]. Similarly, caffeine, a non-selective adenosine receptor antagonist, has been found to reverse the antiplatelet effects of adenosine in vitro [9]. However, chronic caffeine stimulation of human platelets led to an upregulation of adenosine A_2A_ receptors, which correlated with an anti-aggregatory phenotype, elevated cAMP levels, and a reduction in intracellular calcium concentrations [9].

Inflammation and platelet aggregation are interconnected processes that can influence each other in various ways. On the one hand, inflammation can promote platelet aggregation through various mechanisms including cytokines such as interleukin-1 (IL-1) and tumor necrosis factor-alpha (TNF-alpha), which can directly stimulate platelet activation and aggregation. They enhance platelet adhesion to blood vessel walls and increase the expression of adhesion molecules on platelets, facilitating their aggregation. Chemokines can attract and activate platelets. They promote platelet recruitment to the site of inflammation, leading to localized platelet aggregation. Moreover, inflammation can generate reactive oxygen species (ROS). ROS can induce platelet activation and aggregation by causing oxidative stress and triggering platelet-signaling pathways [10].

On the other hand, platelet aggregation can also contribute to inflammation. When platelets aggregate, they release additional inflammatory mediators, such as platelet-derived growth factors and chemokines. These substances can perpetuate and amplify the inflammatory response, leading to a positive feedback loop between platelet aggregation and inflammation [11,12].

It is important to note that while inflammation can promote platelet aggregation, excessive or uncontrolled platelet aggregation can induce thrombosis. The relationship between platelet aggregation and inflammation is particularly important in atherosclerosis, which is usually considered a chronic inflammatory disease. Unstable atherosclerotic plaque rupture, subsequent platelet activation, aggregation, and thrombosis cause stenosis or occlusion of blood vessels, leading to acute cardiovascular disease. Therefore, when developing drug molecules, e.g., for anti-inflammatory treatment, it is necessary to determine their effects on the platelet aggregation process [13].

Cardiovascular diseases have the highest mortality rate of all types of diseases worldwide. Atherosclerosis and thrombotic processes associated with the rupture of vulnerable plaques are the main triggers of cardiovascular and cerebrovascular strokes. Platelets represent the bridge between inflammation and thrombosis, which are fundamental processes in the development of atherothrombosis [14,15,16].

In cardiovascular diseases, oxidative stress and platelet activation are often closely related, and oxidative stress is a factor of great importance for the progression of these diseases [17,18,19]. Thus, the increase in reactive oxygen species (ROS) in the circulation exposes platelets to an activating medium, promoting a change in the platelet phenotype to a pro-adhesive and aggregation-promoting one, which in turn leads to thromboembolic propensity. Therefore, antioxidant compounds could also show the ability to inhibit platelet aggregation induced by various agonists and protect against cardiovascular diseases [20].

The adenosine receptors expressed on platelets could be potential targets for inhibiting platelet activation and for down-regulating inflammation using adenosine A_2A_ agonists [5]. On the other hand, adenosine A_2A_ and A_2B_ receptor antagonists have been shown to exert anti-inflammatory effects [21,22,23], which was also shown in our earlier publications for selected adenosine receptor antagonists [24,25]. For compounds with anti-inflammatory activity, a possible increase in platelet aggregation should be regarded as a potentially serious side effect.

Inhibition of phosphodiesterase (PDE) activity has been suggested to contribute to the clinical efficiency of xanthine derivatives and to account for many of their side effects [26,27,28,29]. PDEs, by catalyzing the hydrolysis of cAMP and cGMP, reduce the intracellular levels of cyclic nucleotides, thus regulating platelet function. Platelets possess three PDE isoforms (PDE2, PDE3, and PDE5) with different selectivity for cAMP and cGMP. The inhibition of PDEs may exert a strong platelet inhibitory effect [30].

In this study, we used different aggregation models to determine the effects of xanthine A_2A_ and A_2B_ adenosine receptor antagonists on platelets as precisely as possible. In addition, we attempted to explain the mechanisms underlying the observed effects. Structurally diverse xanthine derivatives with different activities toward the adenosine receptor subtypes were selected: the selective adenosine A_2A_ receptor antagonists MZ-1497 [31], KD-64 [32], and Tb-42, the moderately selective A_2A_ antagonist TB-46, and the potent and highly selective A_2B_ receptor antagonists PSB-603 [33] and PSB-23066 [34] (Table 1).

## 2. Results

### 2.1. Evaluation of In Vitro Antiplatelet Activity

To evaluate antiplatelet activities of the test compounds, freshly isolated rat whole blood was incubated with compound or vehicle, and the aggregation responses were assessed with a whole blood aggregometer by measuring impedance changes. Platelet aggregation was induced with collagen. First, compounds were screened for anti-aggregation effects at concentration of 10 µM and 100 µM (Figure 1). In this experiment two compounds were able to reduce collagen-induced aggregation at the concentration of 100 µM: the A_2B_ antagonist PSB-603 and the A_2A_ antagonist TB-42, which reduced platelet aggregation by 50.4% and 56.8%, respectively. Compounds KD-64, TB-46, PSB-23066, and MZ-1497 did not affect platelet aggregation.

In the next step, we tested the selected compounds PSB-603 and TB-42 in lower concentrations. PSB-603 prevented platelet aggregation only at the highest tested concentration of 100 µM (by 54.12%) but did not markedly affect platelet activity at lower concentrations (Figure 2). The calculated IC_50_ for PSB-603 was 98.6 ± 27.1 µM (Table 2).

Compound TB-42 exerted a significant antiplatelet effect also in lower concentrations, with an IC_50_ value of 69.3 ± 0.7 µM (Figure 2, Table 2).

As a reference compound, we used the PDE inhibitor 1-methyl-3-isobutylxanthine (IBMX), which effectively inhibited platelet aggregation with an IC_50_ value of 98.8 ± 30.0 µM (Figure 2, Table 2).

In further experiments, aggregation was induced with ADP. In this model of aggregation, we tested the selected compounds as well as IBMX as a positive control. PSB-603, TB-42, and IBMX effectively prevented ADP-induced platelet aggregation (Figure 3), showing higher activity than in the collagen-induced aggregation assay. The calculated IC_50_ values were 82.9 ± 17.0 µM, 58.6 ± 35.6 µM, and 18.5 ± 7.3 µM for PSB-603, TB-42, and IBMX, respectively (Table 2).

### 2.2. Evaluation of Platelet Cytotoxicity

The tested compounds did not cause platelet cytotoxicity. Lactate dehydrogenase activity was determined in platelet-rich plasma incubated with the test compounds as compared to incubation with the medium alone (Figure 4).

### 2.3. Evaluation of Phosphodiesterase (PDE) Inhibition

All the investigated compounds inhibited the activity of PDE2A at low to moderate concentrations. Some of the compounds inhibited PDE3A activity with moderate to high potency, while several compounds displayed low inhibitory effects toward PDE5A (Table 3, Figure 5, Figure 6 and Figure 7). All the studied compounds demonstrated lower inhibition of PDE2A activity than the PDE2A-selective inhibitor *erythro*-9-(2-hydroxy-3-nonyl)adenine (EHNA). MZ-1497, TB-42, and TB-46 exhibited high inhibitory activities against PDE3A with submicromolar IC_50_ values, slightly less potent, but still comparable to the reference compound cilostazol. Conversely, KD-64 demonstrated minimal PDE3A activity inhibition. The A_2B_ receptor antagonists PSB-23066 and PSB-603 did not display any detectable effects against PDE3A activity in the concentrations tested. Similarly, PSB-603, PSB-23066, TB-42, and TB-46 displayed low levels of PDE5A inhibition, much lower than the selective PDE5 inhibitor sildenafil.

### 2.4. Evaluation of In Vitro Lipid Peroxidation (MDA Test)

The xanthine derivative TB-46 showed the greatest antioxidant activity, reaching 53% of the activity of carvedilol at the same concentration of 1000 µM (Table 4). However, at a concentration 10 times lower, the antioxidant activity practically disappeared. The other two compounds active in this test, i.e., TB-42 and MZ-1497, inhibited lipid peroxidation by 38 and 45% of the maximum activity of carvedilol. The other tested compounds showed no activity in the MDA test.

## 3. Discussion

Platelets express adenosine A_2A_ and A_2B_ receptors, whose activation leads to increased intracellular cAMP concentrations, resulting in the inhibition of platelet aggregation [3,4,36]. Therefore, it may be expected that adenosine A_2A_ and A_2B_ receptor antagonists facilitate platelet aggregation, which has been shown in some studies [8,9]. Our present study aimed to evaluate the effects of several xanthine-derived adenosine A_2A_ and A_2B_ receptor antagonists on platelet aggregation. Some of them showed anti-inflammatory activity in our previous studies [24,25], and the exacerbation of platelet aggregation may be a potential side effect.

For aggregation studies, different agonists can be applied individually to identify a pathway potentially affected by the compound tested. Collagen is one of the most important natural platelet agonists. Platelets adhere to the subendothelial matrix through an interaction with von Willebrand factor and glycoprotein GPIb. This reversible adhesion is then followed by firm adhesion mediated by collagen and the major platelet collagen receptors glycoprotein VI and α_2_β_1_ integrin. Stimulation of the platelet GPVI receptor involves tyrosine phosphorylation cascades and leads to activation of PLCγ, phosphoinositide 3-kinases (PI3K), and small G proteins, with subsequently increased intracellular Ca^2+^ concentration. Calcium mobilization is linked to morphologic changes, the exposition of a procoagulant surface, where different coagulation factors are activated, leading to thrombin generation, the secretion of granular content, and the activation of PLA_2_ [36,37,38,39]. In the process of calcium-dependent platelet activation, Ca^2+^ and DAG-regulated guanine nucleotide exchange factor I (CalDAG-GEFI) is a critical protein that acts as a Ca^2+^ sensor, leading to a first wave of activation of the small GTPase Rap1, which subsequently causes a first wave of thromboxane A_2_ (TXA_2_) generation [40]. TXA_2_ is synthesized and activates PKC, and ADP and 5-HT are released. The generation and secretion of these agonists activate additional platelets and amplify platelet activation. These cascades finally lead to the activation of the glycoprotein IIb/IIIa (GPIIb/IIIa) receptor complex, exposure of the binding sites for fibrinogen, and platelet aggregation [38,39,41]. As collagen-induced platelet aggregation involves a cascade of mechanisms, the effects of test compounds that interfere with any of these mechanisms underlying platelet activation and subsequent aggregation may be detectable. Therefore collagen, as a naturally occurring in vivo aggregation activator, provides a general means for assessing platelet function in vitro [42]. The studied compounds did not exacerbate platelet aggregation induced with collagen, and most of them did not influence aggregation, whereas two compounds: PSB-603 and TB-42 even suppressed platelet activity. The determined IC_50_ values for PSB-603 and IBMX were in a similar range of around 100 µM, whereas the IC_50_ for TB-42 was lower. Interestingly, although PSB-603 and PSB-23066 possess similar structures, differing only in the xanthine-*N*1 substituent (propyl versus butyl), PSB-603 showed higher anti-aggregatory activity.

When ADP was used as an aggregation inducer, PSB-603, TB-42, and IBMX again showed marked antiplatelet activity; the determined IC_50_ values were lower than in the collagen-induced aggregation model, most pronouncedly for IBMX. ADP is stored at high concentrations in platelet-dense granules and released upon platelet activation. Released ADP is an essential secondary agonist, which amplifies most of the platelet responses and contributes to the stabilization of the thrombus [43,44]. Aggregation mediated by ADP is mainly due to activation of the G protein-coupled P2Y_1_ and P2Y_12_ receptors, in addition to the ATP-gated P2X_1_ receptor ion channel [45], which facilitates Ca^2+^ influx. The P2Y_1_ receptor is coupled to Gq proteins, which regulate phospholipase C and induce intracellular Ca^2+^ mobilization, leading to aggregation and shape changes [41]. The activation of Gq proteins leads to phospholipase C-β2 (PLCβ_2_) activation and activation of the small G proteins RhoA and Rac with subsequent phosphorylation of kinases belonging to the Src family. PLCβ_2_, downstream of Gq, regulates both inositol trisphosphate (IP_3_) and diacylglycerol (DAG), leading to the release of calcium and activation of protein kinase C (PKC) [6]. The P2Y_12_ receptor is responsible for the amplification of platelet activation initiated by other agonists including ADP. The P2Y_12_ receptor is coupled to G_i_ proteins, leading to the inhibition of adenylate cyclase and the regulation of phosphoinositide-3-kinase. The inhibition of adenylate cyclase leads to a reduction in intracellular cAMP levels, phosphorylation of the serine/threonine protein kinase Akt, stimulation of a second wave of Rap1 activation and TXA_2_ generation, and dephosphorylation of vasodilatory-stimulated phosphoprotein (VASP), resulting in the promotion of the aggregatory response [40,44,45,46]. 

cAMP and cGMP are inhibitory intracellular second messengers controlling platelet activity [47,48,49]. Prostacyclin (PGI_2_) and nitric oxide (NO), which activate adenylate and guanylate cyclases, are potent physiological anti-aggregation factors. An increase in platelet cAMP/cGMP concentrations impinges with platelet activatory signaling pathways, hampering cytoskeletal reorganization, fibrinogen receptor activation, degranulation, and expression of pro-inflammatory mediators. cAMP and cGMP activate protein kinases that phosphorylate specific substrates (i.e., Rap1, VASP), thus hampering receptor/G protein activation, PLC, PKC, and mitogen-activated protein kinase activation, and blocking cytosolic Ca^2+^ elevation [30,50]. Phosphodiesterases catalyze the hydrolysis of cAMP and cGMP into inactive 5′-AMP and 5′-GMP, thereby decreasing the intracellular levels of cyclic nucleotides. Platelets express three PDE isoenzymes: PDE2, PDE3, and PDE5 [51]. PDE2 and PDE3 hydrolyze both cAMP and cGMP, whereas PDE5 specifically hydrolyzes cGMP [30]. The inhibition of PDEs has been shown to exert a strong antiplatelet effect. For example, cilostazol, a specific and strong inhibitor of PDE3 in platelets [52,53], and dipyridamole, an inhibitor of PDE5 and PDE3, both exert strong antiplatelet effects in laboratory animals as well as in humans [30].

As the studied compounds are xanthine derivatives, we considered the possibility that they may inhibit PDEs. Therefore, we assessed their inhibitory effect on PDE activity. As platelets possess PDE2A, PDE3A, and PDE5A as the main isoforms [30], we evaluated the inhibitory potency of the compounds toward PDE2A, PDE3A, and PDE5A. All the studied compounds turned out to be non-selective PDE inhibitors. The inhibitory capacity of the investigated compounds against PDE2A and PDE5A activities was comparable to that of IBMX, a non-selective PDE inhibitor being a xanthine derivative [30,54]. Among the newly investigated xanthines, PSB-603 showing an antiplatelet effect, exerted the strongest inhibitory activity against PDE5A accompanied by inhibitory activity against PDE2A. The second compound with antiplatelet effects, TB-42 showed high inhibitory potency against PDE3A, accompanied by moderate activity against PDE2A as well as PDE5A. Surprisingly, compound TB-46 also exerted strong inhibitory activity against PDE5A and PDE2A and moderate inhibition of PDE3A; however, it did not show an antiplatelet effect. This may be explained by the fact that TB-46 is also the most potent adenosine A_2A_ receptor antagonist among the investigated compounds, which may counteract its antiplatelet effects due to PDE inhibition. It is also possible that other antiplatelet mechanisms are involved in the observed anti-aggregation effect of TB-42.

It should also be noted that the antiplatelet effects of the tested compounds are not due to cytotoxic effects on platelets. We excluded this possibility by measuring the activity of platelet lactate dehydrogenase, which remained unaffected after incubation with the test compounds.

We investigated the effects of xanthine-based A_2A_ and A_2B_ adenosine receptor antagonists on lipid peroxidation. Three compounds: MZ-1497, TB-46, and TB-42 decreased lipid peroxidation, which may potentiate their anti-inflammatory, and in the case of TB-42, also the antiplatelet effects.

For better evaluation of the mechanism of action of the tested compounds, the concentrations of cAMP and cGMP in platelets should be determined in future studies. It should also be noted that the experiments were performed in whole blood, where other blood cells in addition to platelets, are present and may modulate platelet function, e.g., by ATP, ADP, or adenosine release and metabolism. Therefore, further studies are necessary to fully explain the mechanism underlying the antiplatelet effects of the investigated xanthine-based adenosine A_2A_ and A_2B_ receptor antagonists, including studies on purified platelets involving measurements of intraplatelet cAMP/cGMP concentrations.

Summing up, we showed that xanthine-based adenosine A_2A_ and A_2B_ receptor antagonists did not induce but could suppress platelet aggregation induced with collagen, or ADP, respectively (PSB-603 and TB-42). The studied compounds are non-selective PDE inhibitors. TB-42 showed the highest inhibitory activity against PDE3A accompanied by moderate activity against PDE2A and PDE5A. The antiplatelet action of PSB-603 and TB-42 was suggested to be due to inhibition of PDE activity and subsequent increase in cAMP and/or cGMP concentration in platelets. The tested xanthine-based adenosine receptor antagonists did not show any cytotoxic effect on platelets. Thus, the investigated xanthine-based A_2A_ and A_2B_ receptor antagonists are safe with regard to platelet activity. Moreover, our results may provide a basis for the development of multi-target xanthine derivatives for the treatment of inflammation and atherosclerosis, and the prevention of heart infarction and stroke.

## 4. Materials and Methods

### 4.1. Chemicals and Drugs

The materials used included 1-methyl-3-isobutylxanthine (IBMX, Cayman Chemical, Ann Arbor, MI, USA), ADP (Sigma-Aldrich, Hamburg, Germany), and collagen (Hyphen Biomed, Neuville-sur-Oise, France).

Test compounds and reference drugs were dissolved in dimethyl sulfoxide (DMSO) immediately before use. KD-64: (1,3-dimethyl-9-(4-methylcyclohexyl)-pyrimido [2,1-f] purine-2,4-dione), TB-46: (3,7-dimethyl-8-(3-phenylpropoxy)-1-(prop-2-yn-1-yl)--purine-2,6-dione, MZ-1497:8-((6-chloro-2-fluoro-3-methoxybenzyl)amino)-1-ethyl-3,7-dimethyl-purine-2,6-dione, and TB-42: (8-(2-bromobenzyl)oxy)-3,7-dimethyl-1-(prop-2-yn-1-yl)-purine-2,6-dione) were synthesized at the Department of Technology and Biotechnology of Drugs, Faculty of Pharmacy Jagiellonian University Medical College (Poland), and compounds PSB-603 (1-propyl-8-(4-((4-(4-chlorophenyl)piperazin-1-yl)sulfonyl)phenyl)-1*H*-purine-2,6(3*H*,7*H*)-dione) and PSB-23066 (1-butyl-8-(4-((4-(4-chlorophenyl)piperazin-1-yl)sulfonyl)phenyl)-1*H*-purine-2,6(3*H*,7*H*)-dione) were synthesized at the PharmaCenter Bonn, Pharmaceutical Institute, Pharmaceutical & Medicinal Chemistry, Germany, according to previous protocols [31,32,33,34]. Table 1 shows the formulas and affinities for individual subtypes of adenosine receptors.

### 4.2. In Vitro Whole Blood Aggregation Tests

In vitro aggregation tests were conducted using freshly collected whole rat blood with a Multiplate platelet function analyzer (Roche Diagnostic, Mannheim, Germany), the five-channel aggregometer based on measurements of electric impedance, according to previous procedures [55]. Upon activation, platelets adhere and progressively aggregate on a duplicate metal sensor in the analyzer test cuvette. This leads to a change in resistance, which is proportional to the number of platelets adhering to the electrodes.

Blood was drawn from the carotid of rats with hirudin blood tubes (S-Monovette, Hirudin, Sarstedt, Germany). Then, 300 μL of hirudin anticoagulated blood was mixed with 300 μL prewarmed isotonic saline solution containing the studied compound or vehicle (DMSO 0.1%) and preincubated for 3 min at 37 °C with continuous stirring. The agonists (collagen, ADP) were diluted using deionized water. Aggregation was induced by adding collagen (final concentration 1.6 µg/mL) or ADP (final concentration 6.5 µM). The volume of the aggregation inducers did not exceed 2 µL each. Activated platelet function was recorded for 6 min. Multiplate software v. V2.05 was used to analyze the area under the curve (AUC) of the clotting process for each measurement and calculate the mean values.

### 4.3. Impact on Platelet Viability (Cytotoxicity Test)

Cell viability was determined in freshly collected rat platelet-rich plasma (PRP). Whole blood (about 5 mL) was collected from rats into a glass tube containing 0.5 mL of PECT medium (94 nM prostaglandin E1, 0.63 mM Na_2_CO_3_, 90 mM disodium edetate, and 10 mM theophylline).

A density barrier was created by combining 5 mL of 1.320 g/mL 60% iodixanol stock solution (OptiPrep density gradient medium, Sigma-Aldrich, Sant Louis, MO, USA) with 22 mL diluent (0.85% NaCl, 20 mM HEPES-NaOH, pH 7.4, 1 mM disodium edetate). For platelet separation (PRP), 3 mL of each sample was layered over 5 mL of the 1.063 g/mL density barrier. Samples were then centrifuged at 350× *g* for 15 min at 20 °C [56].

The platelets were suspended in Barber’s buffer (0.14 M NaCl, 0.014 M Tris, 10 mM glucose; pH 7.4) [57]. The dilutions were 10× or 20×. The number of platelets (1.5–2.0 × 10^8^/mL) used for the test was measured using a spectrophotometer at λ = 800 nm [58].

The cytotoxic effect of tested compounds on blood platelets was evaluated based on the release of lactate dehydrogenase (LDH), according to the instructions of the kit manufacturer (Cytotoxicity Detection Kit^PLUS^, Cat. No. 04744926001, Merck, Roche, Germany). The time for platelet incubation with the compound was 10 min.

### 4.4. PDE Activity Inhibition Test

The inhibitory activities of the tested and reference compounds were assessed for human recombinant phosphodiesterase (PDE)2A, PDE3A, and PDE5A (SignalChem, Richmond, BC, Canada) by utilizing the PDE-Glo™ catalytic activity assay kit (Promega Corporation, Madison, WI, USA). The test was carried out on 384-well, white, flat-bottom plates, and it adhered to the guidelines provided by the manufacturer with a few alterations. Initially, PDE solutions were diluted in PDE-Glo™ Reaction Buffer and subsequently transferred to the wells at a volume of 6.5 μL. The total amount of PDE2A in the reaction mixture was 3.1 ng, for PDE3A, it was 0.388 ng, and for PDE5A, it was 12.4 ng. The stock solutions of each tested and reference compound were prepared by dissolving the compounds in DMSO to achieve a concentration of 10 mM. Subsequently, the stock solutions were diluted in DMSO, further mixed with the reaction buffer at a *v/v* ratio of 1:5 and transferred to the appropriate wells containing PDEs at a volume of 1 μL. Each inhibitor concentration was tested in triplicate. Then, the plate was incubated for 10 min on a heated plate shaker from Grant Instruments (Cambridge, United Kingdom) at 30 °C. The reaction was initiated by adding 2.5 μL of 0.2 μM cAMP solution (for PDE 3A) or 20 μM cGMP solution (for PDE2A and PDE5A) to the wells. Subsequently, the plate was incubated at 30 °C for 30 min on the heated plate shaker. Control reactions were conducted for each experiment, namely, a no-enzyme negative control without substrate, a positive control containing substrate but no enzyme, and a positive control that included both substrate and PDE but lacked an inhibitor. After the incubation period, the reaction was terminated by adding 2.5 μL of PDE-Glo™ Termination Buffer. Then, PDE-Glo™ Detection Buffer (2.5 μL) was added, followed by a 20 min incubation period at room temperature. The final step involved adding 10 μL of the PDE-Glo Kinase^®^ Reagent to the wells and a subsequent 10 min incubation at room temperature. Luminescence readings were carried out using a POLARstar Omega microplate reader from BMG Labtech (Ortenberg, Germany).

For IC_50_ estimation, the data were represented as a percentage of an uninhibited positive control (that included both substrate and PDE but lacked an inhibitor) and plotted against the inhibitor concentration. IC_50_ values were determined using non-linear regression [59] with ADAPT5 software v. 5.0.63 (BMSR, Los Angeles, CA, USA).

### 4.5. Lipid Peroxidation Assays

The lipid peroxidation was measured by the formation of thiobarbituric acid reactive substances (TBARS) in rat brain homogenate, which was made in 0.9% NaCl containing 10 mg tissue per mL [60]. Briefly, rat brain homogenates (1 mL) were incubated at 37 °C for 5 min with 10 µL of a test compound or vehicle. Lipid peroxidation was initiated with the addition of 50 µL of 0.5 mmol/L FeCl_2_ and 50 µL of 2 mmol/L ascorbic acid. After 30 min of incubation, the reaction was stopped by adding 0.1 mL of 0.2% butylhydroxytoluene. Thiobarbituric acid reagent was then added, and the mixture was heated for 15 min in a boiling water bath. The TBARS was extracted using n-butanol and measured at 532 nm. The amount of TBARS was quantified using a standard curve of malondialdehyde (1,1,3,3-tetraethoxypropane was used as standard).

The compound used to compare the results was carvedilol. Its activity was tested in the concentration range of 10–3000 μM (Figure 8). The reference was the maximum activity of carvedilol at the highest concentration (100%). Test compounds were assayed at concentrations of 100 and 1000 µM. The obtained results were then compared to the activity of the reference compound at identical concentrations (1000 µM).

### 4.6. Statistical Analysis

Data were presented as mean ± standard deviation (SD). Statistical comparisons were made using a one-way analysis of variance (ANOVA), and the significance of the differences between the control group and treated groups was determined using Dunnett’s post hoc test. *p* < 0.05 was considered significant.

## Figures and Tables

**Figure 1 ijms-24-13378-f001:**
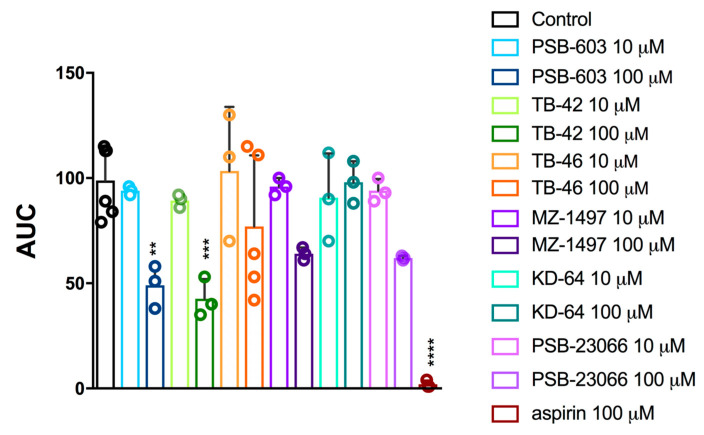
Effects of selected xanthine derivatives (10 and 100 µM) and aspirin (100 µM) on in vitro whole rat blood aggregation induced with collagen (1.6 µg/mL). Results are expressed as mean ± SD, n = 3–6, ** *p* < 0.01, *** *p* < 0.001, **** *p* < 0.0001 vs. control group (0.1% DMSO in saline); one-way ANOVA and Dunnett’s post hoc test. AUC—area under curve.

**Figure 2 ijms-24-13378-f002:**
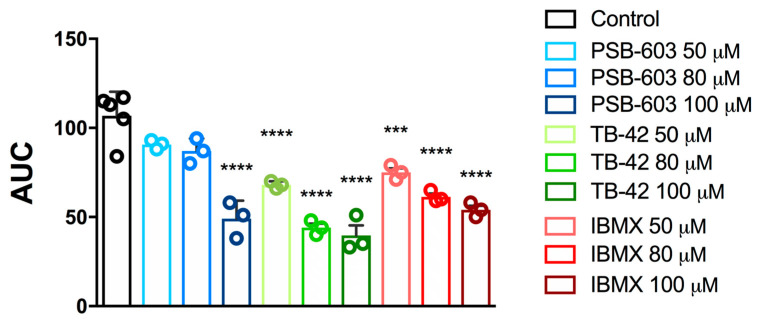
Effects of PSB603, TB42, and IBMX on in vitro whole rat blood aggregation induced with collagen (1.6 µg/mL). Results are expressed as mean ± SD, n = 3-5, *** *p* < 0.001, **** *p* < 0.0001 vs. control group (0.1% DMSO in saline); one-way ANOVA and Dunnett’s post hoc test. AUC—area under curve.

**Figure 3 ijms-24-13378-f003:**
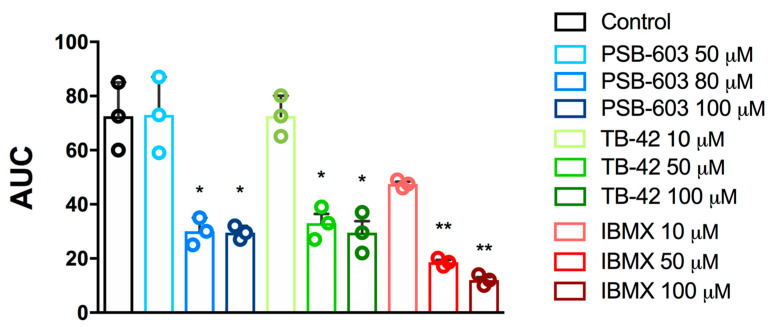
Effects of PSB-603, TB-42, and IBMX on in vitro whole rat blood aggregation induced by ADP (6.5 µM). Results are expressed as mean ± SD, n = 3, * *p* < 0.05, ** *p* < 0.01 versus the control group (0.1% DMSO in saline); one-way ANOVA and Dunnett’s post hoc test. AUC—area under curve.

**Figure 4 ijms-24-13378-f004:**
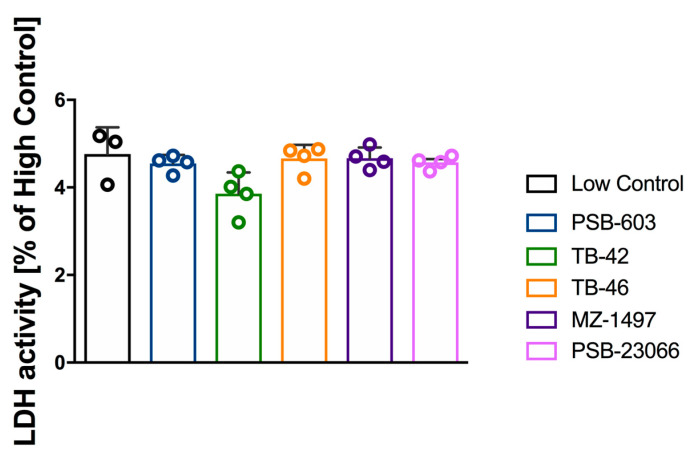
Effects of PSB-603, TB-42, TB-46, MZ-1497, or PSB-23066 on survival of rat blood platelets in vitro. Results are expressed as mean ± SD, n = 3–4; one-way ANOVA. Low Control—only medium (Barber’s buffer), High Control—lysis (medium + lysis solution).

**Figure 5 ijms-24-13378-f005:**
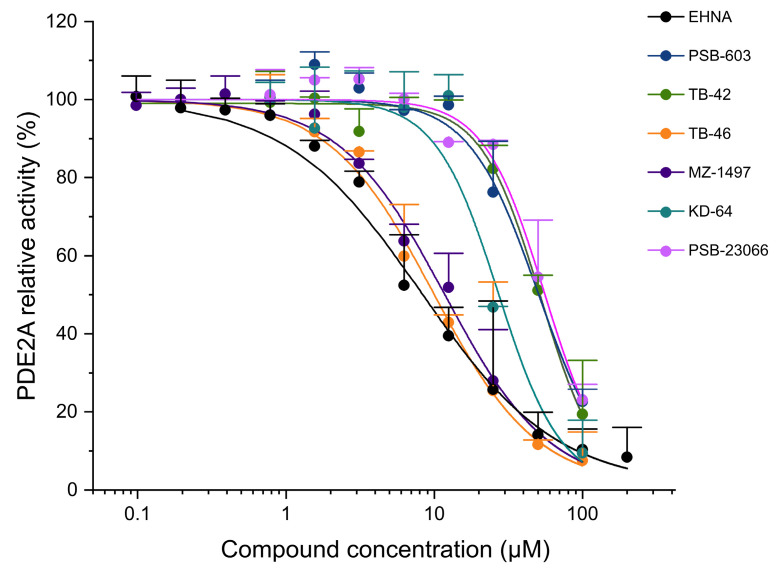
Inhibitory effects of the investigated xanthine derivatives and reference compounds on PDE2A activity. Symbols are the mean (+SD) of three measurements.

**Figure 6 ijms-24-13378-f006:**
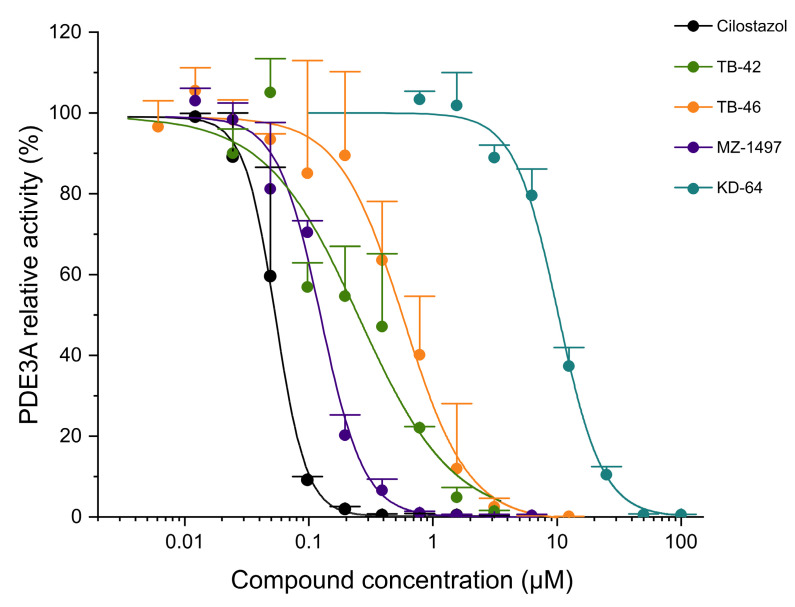
Inhibitory effect of the investigated xanthine derivatives and reference compounds on PDE3A activity. Symbols are the mean (+SD) of three measurements.

**Figure 7 ijms-24-13378-f007:**
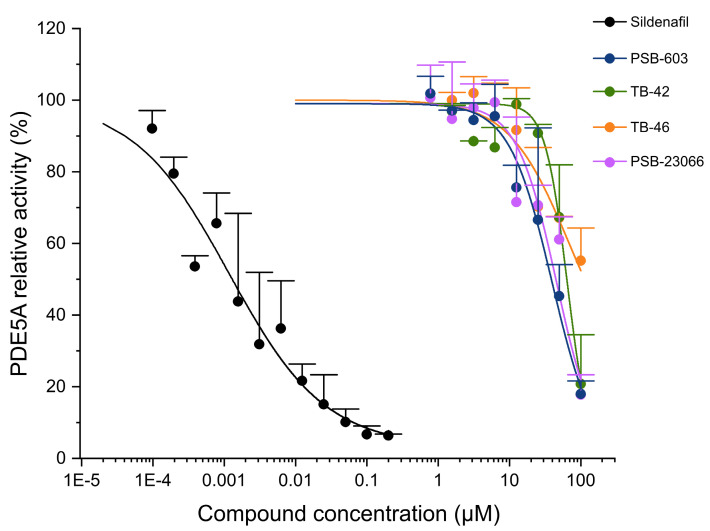
Inhibitory effect of the investigated xanthine derivatives and reference compounds on PDE3A activity. Symbols are the mean (+ SD) of three measurements.

**Figure 8 ijms-24-13378-f008:**
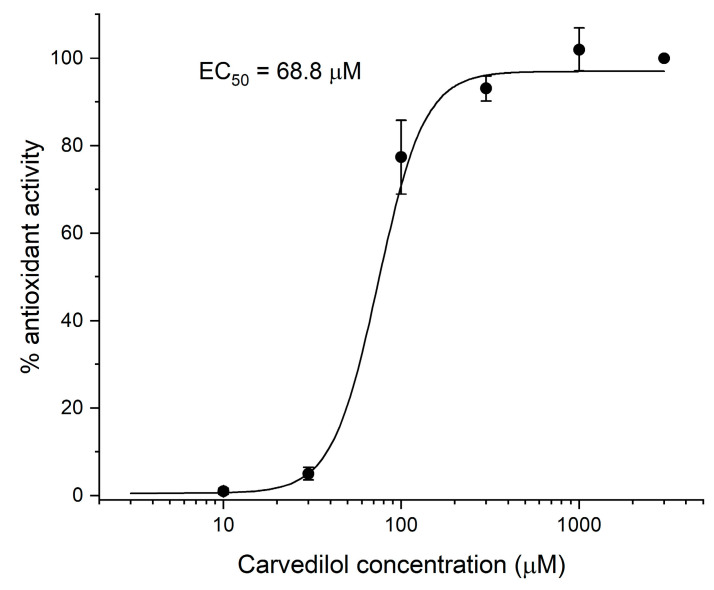
Antioxidant action curve of carvedilol.

**Table 1 ijms-24-13378-t001:** Structures and adenosine receptor affinities of the investigated compounds [31,32,33,34].

**Compound/Structure**	**hA_1_R vs. [^3^H]CCPA**	**hA_2A_R vs. [^3^H]MSX-2**	**hA_2B_R vs. [^3^H]PSB-603**	**hA_3_R vs. [^3^H]PSB-11**
	*K*_i_ ± SEM (nM) (or % inhibition ± SEM at 1 µM)			
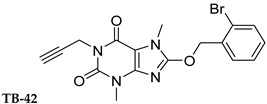	357 ± 49	59.0 ± 10.2	>1000	n.t.
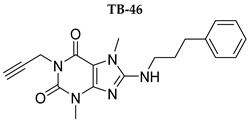	158 ± 12	19.1 ± 2.3	472 ± 174	>1000
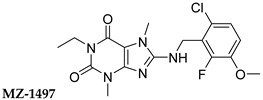	>1000	68.5 ± 15.5	>1000	>1000
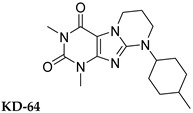	>1000	240	>1000	>1000
**Compound/Structure**	**hA_1_R vs. [^3^H]CCPA**	**hA_2A_R vs. [^3^H]MSX-2**	**hA_2B_R vs. [^3^H]PSB-603**	**hA_3_R vs. [^3^H]PSB-11**
	*K*_i_ ± SEM (nM) (or % inhibition ± SEM at 1 µM)			
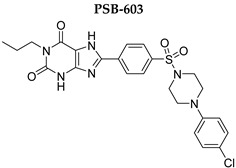	>1000	>1000	0.553 ± 0.103	>1000
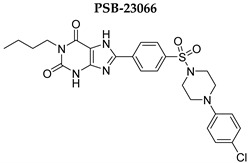	>1000	>1000	0.401 ± 0.136	>1000

n.t.—not tested; h—human.

**Table 2 ijms-24-13378-t002:** Antiplatelet effects of PSB-603, TB-42, and IBMX in an in vitro whole rat blood aggregation assay induced with collagen or ADP.

	IC_50_ ^a^ [μM]	
Compound	Collagen [1.6 µg/mL]	ADP [6.5 μM]
PSB-603	98.6 ± 27.0	82.9 ± 17.0
TB-42	69.3 ± 0.7	58.6 ± 35.6
IBMX	98.8 ± 29.8	18.5 ± 7.3

^a^ IC_50_, concentration inhibiting the aggregation of whole rat blood in vitro by 50%.

**Table 3 ijms-24-13378-t003:** Mean (CV%) ^a^ PDE2A, PDE3A, and PDE5A inhibitory activities of the investigated compounds.

	IC_50_ (CV%) ^a^ (µM)		
Compound	PDE2A	PDE3A	PDE5A
PSB-603	50.1 (23.9)	>100 (n.d.)	39.7 (31.5)
KD-64	27.0 (33.0)	10.3 (7.03)	>100 (n.d.)
PSB-23066	55.3 (36.5)	>100 (n.d.)	45.8 (33.1)
MZ-1497	11.7 (35.9)	0.125 (8.68)	>100 (n.d.)
TB-42	52.1 (16.8)	0.254 (20.7)	64.5 (31.6)
TB-46	9.85 (15.8)	0.580 (20.6)	68.6 (26.6)
EHNA	8.22 (16.3)	-	-
Cilostazol	-	0.0545 (2.39)	-
Sildenafil	-	-	0.00117 (31.0)
IBMX	9.22 [35]	0.71 [35]	45.1 [35]

^a^ CV%, coefficient of variation calculated as CV% = SD/mean × 100; n.d., not determined.

**Table 4 ijms-24-13378-t004:** Ability of test compounds to inhibit lipid peroxidation at a concentration of 1000 μM relative to carvedilol at the same concentration.

Compound	Activity Compared to Carvedilol *
TB-46	59 ± 9.8%
MZ-1497	45 ± 0.7%
TB-42	38 ± 1.1%
PSB-23066	0%
PSB-603	0%
KD-64	0%

* Antioxidant activity expressed as % of carvedilol activity at the same concentration of 1000 µM. Results from two separate experiments in duplicate are presented as the mean ± SD.

## Data Availability

The data presented in this study are available on request from the corresponding author.

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
