# Peer review of "Antiplatelet Effects of Selected Xanthine-Based Adenosine A2A and A2B Receptor Antagonists Determined in Rat Blood"

_ijms, 2023, doi:10.3390/ijms241713378_

Round 1
Reviewer 1 Report
In the original article entitled „Antiplatelet effects of selected xanthine-based adenosine A2A and A2B receptor antagonists” by Dr Kubacka et al., the Authors investigated the effects of a group of xanthine derivatives having antagonistic activity towards adenosine A2A and A2B receptor on platelet aggregation. In my opinion, the paper reports very interesting results and is well-written. Despite there are quite a few papers reporting anti-platelet activity of adenosine receptors agonists alone or in the combination with other anti-platelet drugs, little is known on the anti-platelet action of adenosine receptor antagonists. It implies a novelty of this study. Overall I rate this paper as the very solid work, however I have several comments.
Major comments:
1. The Authors decided to use rat blood for in vitro tests. What was a rationale for this? Performing these experiments using human blood would not be problematic or expensive but undoubtedly more relevant.
2. Are the Authors planning to validate in vitro results with the use of in vivo model(s)? Maybe that was the reason to choose rat blood for in vitro tests? If so, which animal models are considered?
Minor comments:
1. Table 1. Please give units for IC50 values in the table heading (probably uM)
2. Table 1 title (line 158). There is no point to use (A) and (B) in the title of Table 1. Instead, I would recommend to give the concentrations of agonists (ADP and collagen) in the Table heading not in the title.
3. Line 291-292 „It should also be noted that the antiplatelet effects of the tested compounds is not due any cytotoxicity.” The sentence is unclear, please rephrase.
In general, English is decent and uderstandable, however a sligth polishing and proofreading of English would be recommended.
Reviewer 2 Report
This is a review of the manuscript by Kubacka et al. entitled "Antiplatelet effects of selected xanthine-based adenosine A2A and A2B receptor antagonists". This study examines the pharmacological effects of new small molecule xanthine-based adenosine antagonists on platelet aggregation and PDE activity in rat platelets. Methods are appropriate and thorough and statistical analysis is sufficient, but there are some concerns about efficacy of the new compounds. Overall, this is an interesting study that has potentially identified antagonists that do not stimulate platelet activation, which has not been described in the literature to date. Concerns and clarifications are discussed below.
Concerns
Concentrations of the new compounds are too high; 100 mM may be high enough to cause off-target effects, meaning these compounds may be interacting with non-adenosine receptors. Is there any evidence to suggest that other non-adenosine receptors are not being antagonized? Concentrations from 1 – 50 mM should be used with platelets, which may completely eliminate antiplatelet effects observed in Figure 1 and alleviate concerns about off-target inhibition.
In other words, is the platelet inhibition dependent on A2A or A2B? Could other well-known antagonists or knockout mice (they exist) be used to block receptor function independently of the xanthine compounds before platelet aggregation? If the compounds were mostly working through PDE inhibition, then inhibition of aggregation should still occur. Related to this, even though the present compounds inhibit PDE, accumulation of cAMP was not tested. This is important because TB-46 did not show antiplatelet effects, which was explained by its antagonism of receptors. However, the receptors work to raise cAMP, so inhibition of PDE should bypass this effect. Regardless, if the two TB compounds raise cAMP and both inhibit receptors, yet only one inhibits platelets, then other antiplatelet mechanisms may be relevant.
It would be interesting to know if these compounds also inhibit aggregation with purified platelets given the complex environment of whole blood and white blood cells expressing adenosine receptors.
Clarifications
Rats should be mentioned in the title to distinguish from other systems.
The paragraph about cardiovascular disease on line 92 should be earlier in the Introduction, possibly the very first paragraph.
Table 2 should have an "a" after CV in the title similar to Table 1.
Concentrations used in Table 3 are not clear. Line 195 says 1000 mM for the compounds was used, but also states that carvedilol is at the same concentration, which is not confirmed in the Methods on line 415 (3 mM). The concentrations used in Table 3 should be indicated in the table itself. Furthermore, the footnote in this table should have some sort of symbol (e.g. * or whatever the journal prefers) that also appears next to the heading "Activity compared to carvedilol".
Table 4 is not mentioned in Results! Table 4 should be Table 1 in Results with the other tables renumbered.
It is implied on line 257 that Rap1 is only activated P2Y12, but Rap1 is activated more rapidly by a calcium-dependent guanine nucleotide exchange factor (CalDAG GEFI or RasGRP2). Please address this discrepancy and site a CalDAG paper such as Crittenden JR et al Nature Med 2004 or Stefanini L et al Blood 2009.
Formatting
All Tables should be on one page and not spread over two.
Round 2
Reviewer 2 Report
Thank you for addressing my comments and I look forward to future work from your lab.